# Germline *CSF3R* Variant in Chronic Myelomonocytic Leukemia: Linking Genetic Predisposition to Uncommon Hemorrhagic Symptoms

**DOI:** 10.3390/ijms242216021

**Published:** 2023-11-07

**Authors:** Maria Teresa Bochicchio, Giorgia Micucci, Silvia Asioli, Martina Ghetti, Giorgia Simonetti, Alessandro Lucchesi

**Affiliations:** 1Biosciences Laboratory, IRCCS Istituto Romagnolo per lo Studio dei Tumori (IRST) “Dino Amadori”, 47014 Meldola, Italy; teresa.bochicchio@irst.emr.it (M.T.B.); giorgia.simonetti@irst.emr.it (G.S.); 2Hematology Unit, IRCCS Istituto Romagnolo per lo Studio dei Tumori (IRST) “Dino Amadori”, 47014 Meldola, Italy; giorgia.micucci@irst.emr.it; 3Department of Pathology, Morgagni-Pierantoni Hospital, 47121 Forlì, Italy; silvia.asioli@auslromagna.it

**Keywords:** chronic myelomonocytic leukemia, *CSF3R*, bleeding disorders, predisposing genes, NGS

## Abstract

Chronic myelomonocytic leukemia (CMML) is a hematological neoplasm characterized by monocytosis, splenomegaly, thrombocytopenia, and anemia. Moreover, it is associated with *SRSF2* mutations and, rarely, with *CSF3R* variants. We present the case of an 84-year-old patient with persistent anemia and monocytosis. Due to the presence of dysmorphic granulocytes, monocyte atypia, and myeloid precursors in the peripheral blood cells, the patient was subjected to a bone marrow examination. The diagnosis was consistent with CMML type 2. The Hemocoagulative test showed an increase in fibrinolysis markers. Next-generation targeted sequencing showed *TET2* and *SRSF2* mutations, along with an unexpected *CSF3R* germline missense variant, rarely encountered in CMML. The patient started Azacitidine treatment and achieved normal hemostatic process values. In conclusion, we identified a heterozygous germline mutation that, together with *TET2* and *SRSF2* variants, was responsible for the hemorrhagic manifestation.

## 1. Introduction

Chronic myelomonocytic leukemia (CMML) is a myeloid malignancy with overlapping features of myelodysplastic syndrome (MDS) and myeloproliferative neoplasms (MPNs), typically characterized by peripheral monocytosis along with dysplastic features in the bone marrow (BM). Symptoms are usually related to splenomegaly and cytopenias [1]. Bleeding is an unusual event in CMML, and when it does occur, it is generally related to thrombocytopenia. One critical aspect of CMML lies in its WHO classification, which delineates the disease into three subtypes based on the percentage of blasts and promonocytes in bone marrow and peripheral blood. Specifically, CMML-0 is the least aggressive form, characterized by fewer than 5% blasts in the bone marrow and less than 2% in the peripheral blood. CMML-1 escalates in severity with 5–9% bone marrow blasts and 2–4% peripheral blood blasts. CMML-2, the most aggressive subtype, features 10–19% blasts in both compartments and is often associated with the poorest prognosis. This stratification not only assists clinicians in tailoring treatment—from watchful waiting to aggressive chemotherapy and stem cell transplantation—but also serves as a critical factor in patient selection for clinical trials. As such, understanding these subtypes is paramount for both diagnostic precision and the therapeutic strategy in the management of CMML [2]. Molecular alterations often involve epigenetic, splicing, and signaling genes. Mutations occurring in *TET2, SRSF2, ASXL1*, and the RAS signaling pathway genes are common events, with a particularly high incidence when combined with *SRSF2* and *TET2* alterations [3]. Somatic mutations in the genes encoding the colony-stimulating factor 3 receptor (CSF3R) were identified in a minority of CMML patients [4], whereas they were present in more than 80% of patients affected by chronic neutrophilic leukemia (CNL), a rare *BCR*-*ABL1*-negative myeloid malignancy characterized by mature granulocytosis and a high incidence of hemorrhage [5]. Moreover, biallelic germline *CSF3R* nonsense variants are responsible for severe congenital neutropenia (SCN) [6] and have been described in cases that were refractory to G-CSF treatment [7]. Heterozygous *CSF3R* mutations have also been reported as a predisposing condition for the development of lymphoid and myeloid malignancies, such as multiple myeloma and acute lymphoblastic leukemia [8]. Here, we report a case of a CMML patient with an atypical bleeding tendency and a germline *CSF3R* variant.

## 2. Case Presentation

An 84-year-old patient, suffering from autoimmune thyroiditis and atrophic gastritis, reported the appearance of symptoms, including asthenia, hyporexia (a 10% loss of body weight), and swelling of the proximal joints of the hands and ankles. Consequently, she was admitted to the medicine department of a private institution with a suspected autoimmune flare-up. The thyroid-stimulating hormone (TSH) was high, and the esophagogastroduodenoscopy (EGDS) confirmed an associated atrophic gastritis. Blood tests, initially interpretable as an immune-mediated condition, showed persistent normochromic normocytic anemia in the absence of absolute iron or vitamin B12/folic acid deficiency, mild and fluctuating thrombocytopenia (96 − 111 *×* 10^9^/L), and marked monocytosis (the latter progressively worsening from 1.69 to 4.02 × 10^9^/L).

Therapy with low-dose corticosteroids and a change in levothyroxine dosage resulted in a clinical benefit, an improvement in the platelet count (149 × 10^9^/L), and a partial reduction in the monocyte count (1.4 × 10^9^/L). However, a few weeks after being discharged, the patient reported numerous bruises on the lower limbs, and she was referred to our Hematology Unit. The bleeding episodes were major and, specifically, with an ISTH-BAT score 8. With the exception of mucocutaneous signs of bleeding, clinical examination showed nothing remarkable (PFA-EPI and PFA-ADP in the normal range) and, in particular, the patient was negative for splenomegaly or hepatomegaly. 

The blood count at diagnosis was as follows: hemoglobin (Hb) 8.6 g/dL, white blood cells (WBCs) 18,220/mm^3^ (neutrophils 10,750; monocytes 4190, metamyelocytes 2% and myelocytes 6%), and platelets 107,000/mm^3^.

The granulopoiesis was markedly hyperplastic, maturing to dysmorphic polymorphonuclear leukocytes (PMNs), with a diffuse excess of precursors (Figure 1A). An excess of blastic cells was also evident, with a predominance of promonocytic elements amounting to 10–15% of myelopoiesis. Megakaryocytes were increased in number and dysmorphic, erythropoiesis was hyperplastic but with megaloblastoid dyserythropoietic aspects. At histology, an infiltration in nodular aggregates of small lymphocytes—amounting to 20% of the cellularity—with predominantly B phenotype (CD20^+^, BCL2^+^, CD5^−^) was observed at the interstitial level. Mild, diffuse grade 1 WHO-grade marrow fibrosis was also present. The final diagnosis was CMML type 2.

The cytogenetic analysis showed a normal female karyotype (46, XX on 20 metaphases). At the end of the BM aspirate procedure, the patient showed significant bleeding from the biopsy site, thus requiring hospitalization and antifibrinolytic therapy with intravenous and local tranexamic acid. After achieving hemostasis, the patient was discharged but, soon after, reported domestic trauma, and the clinical picture was further complicated by the appearance of hemarthrosis in the right knee. Hemocoagulative tests showed prothrombin time-international normalized ratio (PT-INR) and an activated partial thromboplastin time (aPTT) in the normal range but a marked increase in fibrinolysis markers (D-dimer above detectability ranges) and initial fibrinogen consumption (73 mg/dL). The International Society on Thrombosis and Haemostasis (ISTH) Criteria for Disseminated Intravascular Coagulation (DIC) were not compatible with overt DIC; nonetheless, the patient was admitted to our Hematology Department given the clinical risks.

From a molecular and coagulative point of view, we opted for a better characterization of the disease at its onset. Details on the blood sample collection and analyses are reported in Appendix A.

We performed mutational analysis using next-generation sequencing (NGS, Sophia Myeloid Solution, a 30-gene panel by SOPHiA GENETICS) and we evaluated markers of platelet activation via flow cytometry (in particular, fibrinogen receptor expression, which was determined by procaspase-activating compound-1 (PAC-1) antibody and global coagulation assays (rotational thromboelastometry, ROTEM)).

NGS analysis identified the following variants: a P95L missense mutation in the *SRSF2* gene, a nonsense, and a frameshift *TET2* variant (G1825* and L615Afs*23), and the E808K missense mutation in the *CSF3R* gene. 

The PAC-1 binding capacity was severely impaired (CD61^+^PAC-1^+^ events = 1%). Accordingly, the hemostatic analysis revealed an impaired fibrin polymerization after exposure to tissue factor (TF), with deleterious effects on clot formation in the intrinsic (INTEM) and extrinsic (EXTEM) pathways. In particular, FIBTEM clotting time (CT) was extremely prolonged (3593 s (s). EXTEM data showed a rise in CT (148 s versus a normal range of 38–79 s). In EXTEM, INTEM, and APTEM analysis, clotting formation time (CFT) was increased (318 s, 260 s, and 328 s, respectively), underlying impairment in the initial rate of fibrin polymerization and a decrease in maximum clot firmness (MCF), resulting in a reduced viscoelastic clot strength (Figure 1B). 

Azacitidine (AZA) treatment was started at the standard dosage of 75 mg/mq for 7 days every 28 days. Treatment was well tolerated and led to an improvement in the hematologic parameters, as well as on the coagulative tests, which was consistent with the resolution of the bleeding complications. 

During treatment, all values, including global coagulation tests, showed a progressive shift towards a normal hemostatic process (Figure 1C). 

To characterize the patient’s mutational landscape, targeted NGS resequencing was performed at diagnosis and after the first and ninth AZA cycles on BM and/or peripheral blood (PB) samples (Table 1). 

The allelic ratio (AR) of *TET2* and *SRSF2* mutations was significantly reduced in both the BM and PB samples after nine AZA cycles, while the variant allele frequency (VAF) of *CSF3R* E808K was 50% at all the time points. To validate *CSF3R* mutation and to test the hypothesis of a germline origin of this variant, we performed Sanger sequencing of *CSF3R* exon 17 on DNA isolated from all the samples and from saliva and blood CD3^+^ and CD3^−^ cells as control, confirming the presence of a heterozygous germline variant (Figure 2). The main clinical and laboratory information is summarized in Figure 1D. 

## 3. Discussion

CMML is a clonal hematopoietic disease commonly associated with the presence of mutations in genes involved in splicing regulation and epigenetic and proliferation control [9]. Somatic mutations of the *CSF3R* gene have also been reported in small secondary clones [10]. At the same time, *CSF3R* mutations have been widely reported in CNL [5,11], which shares some morphological clinical and molecular traits with CMML [12]. *CSF3R* mutations have been included in the CNL diagnostic criteria according to the WHO classification. Specifically, the somatic T618I mutation is a hallmark of CNL and atypical chronic myeloid leukemia (aCML) [13], whereas, to date, about twenty cases of CMML carrying T618I have been reported [10,12]. T618I localizes in the fibronectin-like type III domain and has a well-known oncogenic potential [11,14,15].

The *CSF3R* encodes the receptor for the colony-stimulating factor (G-CSF), a cytokine that controls the production, proliferation, and differentiation of granulocytes [16]. Acquired *CSF3R* mutations are responsible for SCN, which is considered a preleukemic bone marrow failure syndrome with an approximately 20% risk of developing acute myeloid leukemia (AML) or MDS, especially when they occur in association with mutations targeting other genes such as *RUNX1*. These mutations usually appear early in the disease history and drive leukemogenesis [17].

Our patient harbored the E808K, a well-known mutation, which has been previously described as both somatic and germline in patients affected by myeloid malignancies, particularly MDS or MDS/MPN. The E808K mutation lies within the cytoplasmic domain of the *CSF3R* protein and is described as a predisposing leukemia variant since the majority of patients develop AML [18]. Functional studies showed that E808K results in decreased colony formation compared to the wild-type protein in cultured cells with no MAP kinases activation [19] and no transformation potential [20]. 

Therefore, the germline heterozygous *CSF3R* mutation detected in our patient could mean there was a predisposition to developing CMML, in combination with the acquisition of somatic mutations within the *TET2* and *SRSF2* genes, which are commonly detected in this myeloid malignancy [9]. The presence of the germline *CSF3R* variant was not pathogenetic per se, as the patient had neither a personal history of hematological abnormalities nor neutrophilia or neutropenia. 

From a clinical point of view, our patient showed a normal bleeding time, and her platelet count only decreased slightly. However, her clinical behavior at diagnosis was characterized by a severe bleeding tendency, which is unusual in CMML cases and is generally related to thrombocytopenia. Conversely, hemorrhagic diathesis is frequent in CNL, presenting several cases with cerebral hemorrhage despite the platelet count. Bleeding in CNL has been attributed to fibrinogen consumption, perhaps because of the paraneoplastic process, a platelet dysfunction [21], or vascular leukocytic infiltration [22]. The patient’s coagulation parameter analysis demonstrated that the hemorrhages were the result of a fibrinogen impairment with prolonged CT and CFT. This showed an underlying impairment in the initial rate of fibrin polymerization. Moreover, it was clear that the viscoelastic strength of the clot was reduced. The hemorrhagic tendency in *CSF3R*-mutated CNL led us to speculate that the E808K germline *CSF3R* mutation could have a role in the hemorrhagic phenotype observed in our CMML patient, who did not present a critical value of thrombocytopenia to justify her bleeding manifestations.

## 4. Conclusions

In conclusion, we identified a heterozygous germline mutation that could have a predisposition to developing CMML, which is likely after the acquisition of *TET2* and *SRSF2* mutations as definitive leukemia drivers, and to a paraneoplastic hyperfibrinolysis that could have been responsible for the severe hemorrhagic clinical manifestations. Further efforts are needed to better understand the role of *CSF3R* mutations in inducing bleeding.

## Figures and Tables

**Figure 1 ijms-24-16021-f001:**
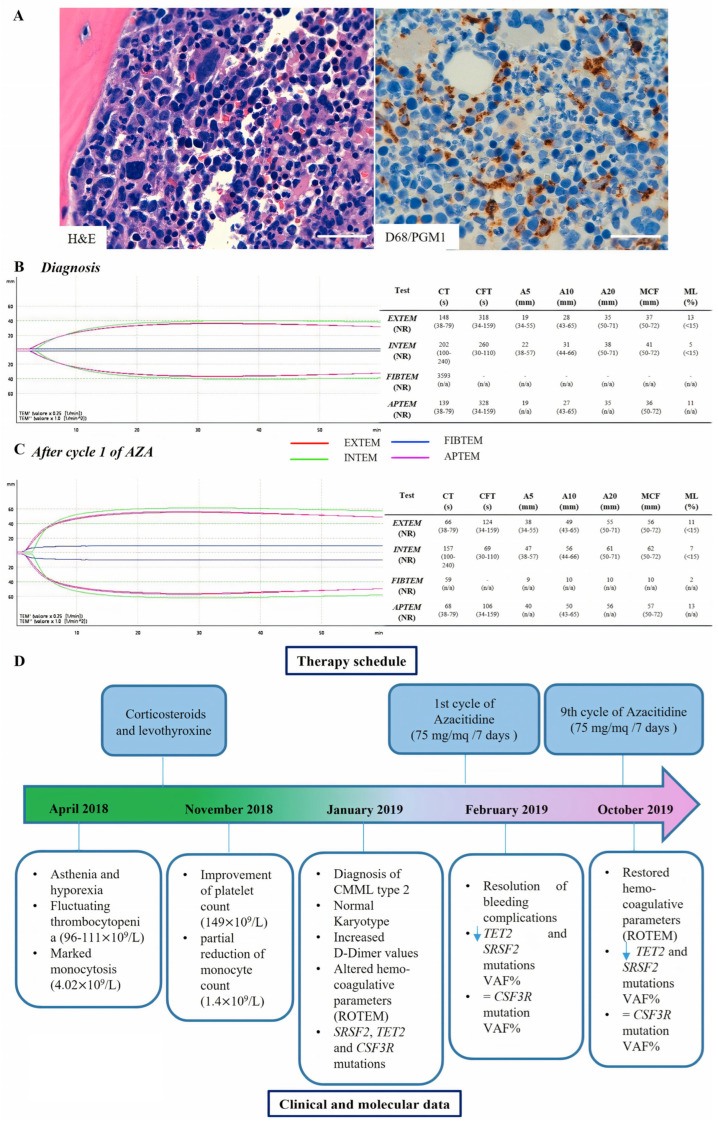
(**A**) Morphological characterization of the BM biopsy. Hematoxylin and eosin (H&E, scale bar 5 µm and immunohistochemistry (IHC) stains for CD68 (CD68/PGM1, scale bar 5 µm) showing hypercellular bone marrow with prominent granulopoiesis, atypical megakaryocytes, and increased CD68/PGM1-positive monocytes. Analysis of coagulation parameters via ROTEM at diagnosis (**B**) and after cycle one of Azacitidine treatment (**C**). Units and reference values are displayed in brackets (Clotting Time: CT, Clot Formation Time: CFT, Amplitude five minutes after CT: A5, Amplitude ten minutes after CT: A10, Amplitude 20 min after CT: A20, Maximum Clot Firmness: MCF, Maximum Lysis: ML, Normality Range: NR). (**D**) Timeline displaying the therapy schedule on the upper part and the clinical history and molecular data on the bottom part. The light blue arrows mean a decrease of VAF% of *TET2* and *SRSF2* genes.

**Figure 2 ijms-24-16021-f002:**
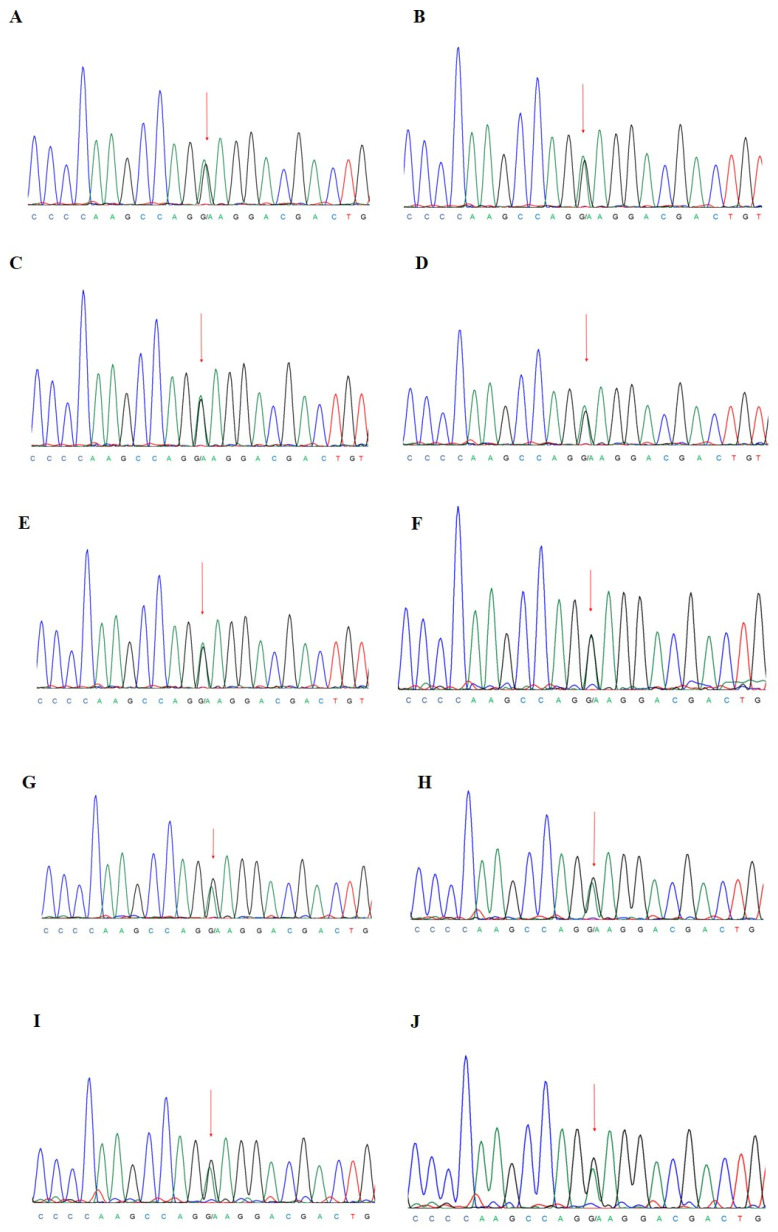
Sanger sequencing validation of *CSF3R* exon 17 E808K missense mutation at diagnosis on BM and PB mononuclear cells (**A**,**B**), after 1 cycle of AZA on PB mononuclear cells (**C**), after 4 cycles of AZA on BM and PB mononuclear cells (**D**,**E**), on saliva (**F**), on PB and BM CD3^+^ cells (**G**,**H**), and on PB and BM CD3^−^ cells (**I**,**J**). Red arrows show the G/A substitution. Blue peak is cytosine, green peak is adenine, black peak is guanine and red peak is thymine.

**Table 1 ijms-24-16021-t001:** Mutational profile of the bone marrow and peripheral blood samples at diagnosis and after the first and ninth Azacitidine cycles.

	Tissue	Gene	Exon	Type	Coding	Amino Acid Change	VAF (%)	Coding Consequence
**DIAGNOSIS**	BM	*TET2*	11	SNP	c.5473C > T	p.(Gln1825*)	41.7	nonsense
		*TET2*	3	INDEL	c.1842dupG	p.(Leu615Alafs*23)	41.5	frameshift
		*CSF3R*	17	SNP	c.2422G > A	p.(Glu808Ly)	49.5	missense
		*SRSF2*	1	SNP	c.284C > T	p.(Pro95Leu)	43	missense
	*PB*	*TET2*	3	INDEL	c.1842dupG	p.(Leu615Alafs*23)	37.7	frameshift
		*TET2*	11	SNP	c.5473C > T	p.(Gln1825*)	36.8	nonsense
		*CSF3R*	17	SNP	c.2422G > A	p.(Glu808Ly)	47.7	missense
		*SRSF2*	1	SNP	c.284C > T	p.(Pro95Leu)	35.9	missense
**POST 1 AZA CYCLE**	*PB*	*TET2*	3	INDEL	c.1842dupG	p.(Leu615Alafs*23)	37.7	frameshift
		*TET2*	11	SNP	c.5473C > T	p.(Gln1825*)	41.6	nonsense
		*CSF3R*	17	SNP	c.2422G > A	p.(Glu808Ly)	49.8	missense
		*SRSF2*	1	SNP	c.284C > T	p.(Pro95Leu)	39.3	missense
**POST 9 AZA CYCLES**	*BM*	*TET2*	11	SNP	c.5473C > T	p.(Gln1825*)	18.4	nonsense
		*TET2*	3	INDEL	c.1842dupG	p.(Leu615Alafs*23)	20.2	frameshift
		*CSF3R*	17	SNP	c.2422G > A	p.(Glu808Lys)	49.9	missense
		*SRSF2*	1	SNP	c.284C > T	p.(Pro95Leu)	20.5	missense
	*PB*	*TET2*	3	INDEL	c.1842dupG	p.(Leu615Alafs*23)	28.6	frameshift
		*TET2*	11	SNP	c.5473C > T	p.(Gln1825*)	29.9	nonsense
		*CSF3R*	17	SNP	c.2422G > A	p.(Glu808Lys)	49.5	missense
		*SRSF2*	1	SNP	c.284C > T	p.(Pro95Leu)	27.3	missense

## Data Availability

The data supporting the findings of this study are available within the article (and in its Appendix A).

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
