# Peer review of "Germline CSF3R Variant in Chronic Myelomonocytic Leukemia: Linking Genetic Predisposition to Uncommon Hemorrhagic Symptoms"

_ijms, 2023, doi:10.3390/ijms242216021_

Round 1

Reviewer 1 Report

Comments and Suggestions for Authors

The Authors present an interesting case report about CMML and bleeding tendency.

I ha ve a few questions/obeservation:

- The bleeding episodes were Major or simply CRNMB according to ISTH? This information should be present in the paper.

- Were other platelet functional tests performed other than the TEG/ROTEM mentioned?

Comments on the Quality of English Language

- In the paper the patient is consistently presented as female with a 46 XX karyotype, but in line 176 is referred as "him".  

Author Response

Response to reviewers

The authors would like to thank the reviewers and the editor for their suggestions allowing us to improve the case report. 

Reviewer #1

- The bleeding episodes were Major or simply CRNMB according to ISTH? This information should be present in the paper.

The bleeding episodes were major and specifically, ISTH-BAT score 8. We added this information in the paper (page 3 line 95-96).

- Were other platelet functional tests performed other than the TEG/ROTEM mentioned?

Yes, other platelet functional tests were performed using PFA-100. Specifically PFA-EPI 135 seconds (range 89-165) and PFA-ADP 84 seconds (range 71-118). We added this information in the paper (page 3 line 97-98).

- In the paper the patient is consistently presented as female with a 46 XX karyotype, but in line 176 is referred as "him".  

We corrected him with her. 

Reviewer 2 Report

Comments and Suggestions for Authors

Bochicchio et. al presented an interesting case report study about G-CSF3R germline mutation and its association with abnormal bleeding disorder in CMML patients. The mutation is rare in CMML patients, and there is limited knowledge about its pathogenesis or prognosis towards disease progression. This report adds potential value to this knowledge, which can assist in future diagnosis.

However, there are some minor concerns that need to be addressed before final approval to publish this report. Firstly, the author should describe CMMS type 2 in the introduction. Secondly, in the Case presentation, the author mentioned autoimmune thyroiditis, but it is unclear how the author ruled out this pre-existing chronic disorder's connection with the abnormal bleeding parameter. Thirdly, Aza treatment resolved the bleeding disorder, but interestingly the VAF remained unchanged until after the 9th cycle of Aza treatment. Therefore, the authors need to describe this discrepancy if they consider the heterozygous mutation of CSF3-R to be the primary manifestation of this abnormal bleeding parameter. Finally, the authors should incorporate the sequencing figure (from Supply) into the main case report, with proper marking and figure legends.

Comments on the Quality of English Language

Minor checking is needded.

Author Response

Response to reviewers

The authors would like to thank the reviewers and the editor for their suggestions allowing us to improve the case report.

Reviewer #2

Bochicchio et. al presented an interesting case report study about G-CSF3R germline mutation and its association with abnormal bleeding disorder in CMML patients. The mutation is rare in CMML patients, and there is limited knowledge about its pathogenesis or prognosis towards disease progression. This report adds potential value to this knowledge, which can assist in future diagnosis.

We thank the reviewer 2 for his/her appreciation.

However, there are some minor concerns that need to be addressed before final approval to publish this report. 

-Firstly, the author should describe CMML type 2 in the introduction. 

We have added in the introduction part the description of CMML type 2.

“One critical aspect of CMML lies in its WHO classification, which delineates the disease into three subtypes based on the percentage of blasts and promonocytes in bone marrow and peripheral blood. Specifically, CMML-0 is the least aggressive form, characterized by fewer than 5% blasts in the bone marrow and less than 2% in the peripheral blood. CMML-1 escalates in severity with 5-9% bone marrow blasts and 2-4% peripheral blood blasts. CMML-2, the most aggressive subtype, features 10-19% blasts in both compartments and is often associated with the poorest prognosis. This stratification not only assists clinicians in tailoring treatment—from watchful waiting to aggressive chemotherapy and stem cell transplantation—but also serves as a critical factor in patient selection for clinical trials. As such, understanding these subtypes is paramount for both diagnostic precision and therapeutic strategy in the management of CMML.” We added this information in the paper (page 2 line 45-57).

- Secondly, in the Case presentation, the author mentioned autoimmune thyroiditis, but it is unclear how the author ruled out this pre-existing chronic disorder's connection with the abnormal bleeding parameter. 

Thyroiditis is rarely associated with acquired hemophilia. Our case presented an evident deficit in fibrin polymerization that arose together with the disease.

-Thirdly, Aza treatment resolved the bleeding disorder, but interestingly the VAF remained unchanged until after the 9th cycle of Aza treatment. Therefore, the authors need to describe this discrepancy if they consider the heterozygous mutation of CSF3R to be the primary manifestation of this abnormal bleeding parameter. 

The heterozygous CSF3R mutation was of germline origin. Its VAF remained unchanged, however Aza treatment improved the symptoms.

-Finally, the authors should incorporate the sequencing figure (from Supply) into the main case report, with proper marking and figure legends. 

We incorporated the sequencing figure in the main case report as Figure 2 with the figure legend.

Reviewer 3 Report

Comments and Suggestions for Authors

The authors describe the interesting case of a CMML patient with a germline mutation in CSF3R and alteration of the blood clotting. The report is interesting but should be more carefully put in the context of the recent literature on the subject.

Page 1,

line 34-35: is not necessary to specify "thrombocytopenia and anemia" when cytopenias are mentioned.

line 35: the sentence on bleeding is best to be merged with the previous one. Otherwise, change the sentence on molecular alteration as "Molecular aberration IN CMML ..."

Page 2

lines 61-63: It is not necessary to specify the (negative) radiological findings. please delete the sentence.

lines 63-70: the clinical, histopathological and cytomorphological descriptions of the patient should be much improved.

Please mention the value of Hb, WCB (tot), and neutrophils at CBC. Was splenomegaly present/absent?

Describe the features of the bone marrow biopsy (at least cellularity, dysplastic lineage(s), % of monocytes/promono (CD14) and % of blasts.

Describe the dysplastic features of granulocytes at the BM aspirate examination (this is a very important differential feature with aCML and CNL in a CSF3R-mutate patient).

Lines 72-73 The sentence is not clear, please revise it.

Page7-discussion

Given the much more frequent association of CSF3R mutation and a clinical-pathological picture of either CNL or aCML, the authors should discuss this differential diagnosis, in line with the recent literature (two quite relevant contributions on the topic were not mentioned: Palomo L, et al. Blood. 2020;136(16):1851-1862. doi:10.1182/blood.2019004229 AND Guastafierro V, et al. Leuk Lymphoma. 2023;64(9):1566-1573. doi:10.1080/10428194.2023.2227750) Please include such articles in the reference list and use it to deepener the discussion concerning the molecular differential diagnosis between CMML, CNL and aCML.

Moreover, the authors should list and explain the molecular difference between CSF3R mutations e.g. T861I vs Y752X. The patient's E808K mutation, is very close to the region of second example and may likely have the same effect. 

Table 1: The table is very large and mostly useless concerning the main points of the article. Please either depict the changes in VAF% with a figure or simply delete the table.

Figure 1A: the BM pictures are of poor quality and depict the same detail at different magnification. Change the PGM1 picture (non informative and actually depicting a normal/low number of resident macrophages) with a CD14 and change the 40x magnification with a picture depicting the granulocytic atypia (smear). 

Comments on the Quality of English Language

English language should be improved by an English speaker person.

Author Response

Response to reviewers

The authors would like to thank the reviewers and the editor for their suggestions allowing us to improve the case report.

Reviewer #3

The authors describe the interesting case of a CMML patient with a germline mutation in CSF3R and alteration of the blood clotting. The report is interesting but should be more carefully put in the context of the recent literature on the subject.

Page 1,

line 34-35: is not necessary to specify "thrombocytopenia and anemia" when cytopenias are mentioned.

line 35: the sentence on bleeding is best to be merged with the previous one. Otherwise, change the sentence on molecular alteration as "Molecular aberration IN CMML ..."

We followed these suggestions and corrected the text in the introduction part. 

Page 2

lines 61-63: It is not necessary to specify the (negative) radiological findings. please delete the sentence.

We removed the sentence.

lines 63-70: the clinical, histopathological and cytomorphological descriptions of the patient should be much improved.

We added the following description in order to better describe clinical, histopathological and cytomorphological patient features.

Please mention the value of Hb, WCB (tot), and neutrophils at CBC. Was splenomegaly present/absent? 

The blood count at the diagnosis was as follows: hemoglobin (Hb) 8.6 g/dL, white blood cells (WBC) 18,220/mm^3 (neutrophils 10,750; monocytes 4190, metamyelocytes 2% and myelocytes 6%), platelets 107,000/mm^3. Clinical examination showed nothing remarkable, and in particular the patient was negative for splenomegaly or hepatomegaly.

Describe the features of the bone marrow biopsy (at least cellularity, dysplastic lineage(s), % of monocytes/promono (CD14) and % of blasts. 

Describe the dysplastic features of granulocytes at the BM aspirate examination (this is a very important differential feature with aCML and CNL in a CSF3R-mutate patient).

The granulopoiesis was markedly hyperplastic, maturing to dysmorphic polymorphonuclear leukocytes (PMNs), with diffuse excess of precursors. An excess of blastic cells was also evident, with predominance of promonocytic elements amounting to 10-15% of myelopoiesis. Megakaryocytes were increased in number and dysmorphic, erythropoiesis was hyperplastic, but with megaloblastoid dyserythropoietic aspects. At histology, an infiltration in nodular aggregates of small lymphocytes - amounting to 20% of the cellularity - with predominantly B phenotype (CD20+, BCL2+, CD5-) was observed at the intestitial level. Mild, diffuse grade 1 WHO-grade marrow fibrosis was also present.

Lines 72-73 The sentence is not clear, please revise it. 

We clarified the sentence.

Page7-discussion

Given the much more frequent association of CSF3R mutation and a clinical-pathological picture of either CNL or aCML, the authors should discuss this differential diagnosis, in line with the recent literature (two quite relevant contributions on the topic were not mentioned: Palomo L, et al. Blood. 2020;136(16):1851-1862. doi:10.1182/blood.2019004229 and Guastafierro V, et al. Leuk Lymphoma. 2023;64(9):1566-1573. doi:10.1080/10428194.2023.2227750) Please include such articles in the reference list and use it to deepener the discussion concerning the molecular differential diagnosis between CMML, CNL and aCML.

Moreover, the authors should list and explain the molecular difference between CSF3R mutations e.g. T861I vs Y752X. The patient's E808K mutation, is very close to the region of second example and may likely have the same effect. 

We thank the reviewer for the input. We have modified the discussion accordingly and we have added new references. Page 12 lines 216-245.

Table 1: The table is very large and mostly useless concerning the main points of the article. Please either depict the changes in VAF% with a figure or simply delete the table. 

We here propose a simplified version of the table just showing the most relevant information (Table 1). 

 Figure 1A: the BM pictures are of poor quality and depict the same detail at different magnification. Change the PGM1 picture (non informative and actually depicting a normal/low number of resident macrophages) with a CD14 and change the 40x magnification with a picture depicting the granulocytic atypia (smear). 

We removed the 40x magnification  picture as suggested. Unfortunately the CD14 staining could not be performed. Therefore, if the reviewer agrees, we suggest to keep  the CD68/PGM1 figure, since it shows the increased number of monocytes. 

English language should be improved by an English speaker person.

We improved english language by an English speaker person.